# Successful Application of Team Resource Management in Scrub Typhus Infection with Septic Shock

**DOI:** 10.3390/ijerph191710683

**Published:** 2022-08-27

**Authors:** I-Hung Chen, Cher-Min Fong, Hsing-Hua Stella Chang, Jen-Hsien Lin

**Affiliations:** 1Department of Internal Medicine, Pingtung Branch of Kaohsiung Armed Forces General Hospital, Pingtung 90092, Taiwan; 2Department of Business Management, National Sun Yat-Sen University, Kaohsiung 80424, Taiwan; 3Institute of Medical Science and Technology, National Sun Yat-Sen University, Kaohsiung 80424, Taiwan; 4International Master of Business Administration, National Taichung University of Education, Taichung 40359, Taiwan; 5Department of Internal Medicine, Kaohsiung Armed Forces General Hospital, Kaohsiung 80284, Taiwan

**Keywords:** scrub typhus, team resource management conference, family members, Weil-Felix test

## Abstract

The fatality rate of scrub typhus infection with septic shock is quite high if timely and correct diagnosis and treatment are not obtained. There are few studies in the literature on the subject of holding TRM conferences to discuss the condition and reach a consensus on treatment. A TRM conference has the significance of early intervention by the medical team and consensus on therapy from the medical doctors and family members. We report the case of scrub typhus infection with septic shock. On the day the patient was hospitalized, the medical team held a TRM conference and invited family members to attend. We found that the eschar on the patient may be related to scrub typhus, which was later confirmed by a positive Weil-Felix test and PCR analysis. Under the consensus treatment, the patient’s condition improved considerably within the next day. The most significant difference between the TRM conference and the clinical specialist’s consultation is that it can quickly narrow the cognitive gap between doctors and family members and reach a consensus on the patient’s therapy strategy, truly avoid medical disputes, and effectively share the stress of attending physician. In this case report, we highlight the significance of the TRM conference.

## 1. Introduction

Scrub typhus is a rickettsia disease caused by *Orientia tsutsugamushi* [1]. This disease is endemic in many countries, including Taiwan, China, Japan, Korea, and other southeastern Asia with a serious public health problem resulting in disease in one million people each year [2]. A diagnosis of scrub typhus could be confirmed by the positive result of immunofluorescent antibodies and polymerase chain reaction (PCR) analysis [2]. Doxycycline is the medicine of choice [3]. Without appropriate treatment, scrub typhus can result in severe multiorgan failure with a case fatality rate up to 70% [2]. Septic shock, acute respiratory failure, acute kidney injury, disseminated intravascular coagulation (DIC), and multiorgan failure are the causes of death [4]. We reported a case of scrub typhus with septic shock, multiorgan dysfunction, DIC, and cardiac arrhythmia of atrial fibrillation with rapid ventricular rate. The CHA2DS2-VASc score is recommended to assess thromboembolic risk and to guide antithrombotic therapy in patients with atrial fibrillation or flutter [5]. Multidisciplinary support and expertise can effectively help clinical teams in reaching their full potential. This is the spirit of the team resource management (TRM) conference [6]. A TRM conference has the significance of early intervention by the medical team and consensus on therapy from the medical doctors and family members. Family members were also invited to attend the conference which is the significant difference from the clinical specialist’s consultation. This patient was fortunately discharged without any clinical sequela. Successful therapy was guided under strategic application of a TRM conference held as soon as possible following with early medical team’s intervention and consensus of therapy from doctors and family members.

## 2. Case Report

A 60-year-old man, also a physician, was admitted with fever, general soreness, and headache that had developed 3 days before admission. He had a travel history of mountain climbing in the past 10 days. Since the clinical condition gradually deteriorated with fever and shortness of breath, he was sent to ER by his family in the early morning. On arrival at our ER, he was lucid but had generalized cold sweating. His blood pressure was 121/77 mmHg, body temperature was 40 °C, pulse rate was 103 beats/min, and respiratory rate was 24 times/min. There were no eschars noted initially. His heart sounds were normal. A coarse breathing sound was heard over both lung fields. His abdomen was soft without tenderness.

Laboratory test results showed the following: white cell count, 8.26 × 10^3^/uL; hemoglobin, 14.2 g/dL; platelet count, 15.8 × 10^3^/uL; blood urea nitrogen (BUN), 20 mg/dL; creatinine (Cr), 1.5 mg/dL; albumin, 3.6 g/dL; aspartate aminotransferase, 100 U/L; alanine aminotransferase, 54 U/L; bilirubin total, 0.59/ mg/dL; alkaline phosphatase, 28 U/L; lactate dehydrogenase, 503 U/L; CRP 14.63 mg/dL; Routine urinalysis showed proteinuria: 2+; 0-5 WBC cells/HPF,3-5 RBC cells/HPF and no cast. DIC profiles disclosed: prothrombin time 10.1 s (12 s of control), partial thromboplastin time 32.5 s (26.9 s of control), fibrinogen 329.3 g/L (normal: 2.16–3.18 g/L), fibrin degradation products (FDP) > 40 mg/L (normal: 0–10 mg/L), D-dimer > g/L (normal: < 500 g/L). Dengue Virus Ab IgM (-), Dengue virus Ab IgG (-), Dengue NS1 Ag test (-), Influenza A Ag(-), Influenza B Ag(-). The venous blood gas analysis revealed pH 7.531, PaO2, 49.9 mmHg, PaCO2, 31.7 mmHg, HCO3, 28.8 meq/L, chest X-ray film showed emphysema of both lungs, 12 lead EKG showed atrial fibrillation with rapid ventricular rate. Echocardiography revealed normal regional wall motion with preserved LV (left ventricle) systolic function, mild MR (mitral regurgitation), mild TR (tricuspid regurgitation), mild PR (pulmonary regurgitation), without valvular vegetation, and atrial fibrillation with rapid ventricular response during the study. Abdominal sonography revealed chronic parenchymal liver disease; the pancreas and spleen appeared normal.

Since the relative unstable vital sign included atrial fibrillation with rapid ventricular rate, shortness of breath with increased respiratory rate, and decreased blood pressure, he was then admitted to the intensive care unit, with oxygen supplied through a nasal cannula. The impression was septic shock with multiorgan dysfunction (acute kidney and liver injury, and DIC), cause to be determined. Shortly after the patient was admitted to the ICU, a professional team resource management (TRM) conference was held by the director of the medical department, patient’s wife and nine clinical specialists were invited, including an infectious disease physician, a nephrologist, a hepatobiliary physician, a cardiologist, a pulmonologist, dermatologists, neurologists, ICU attending physicians and chief nurse.

Due to the obvious travel history, another thorough physical examination was performed on the patient, which revealed erythema on the right buttock, with a central skin lesion forming a black crust over the ulcer (Figure 1). After thorough discussion, first, the skin lesion had been confirmed as an eschar, the specific feature of a wound due to a mite bite was highly suspected by a dermatologist. Scrub typhus infection with septic shock was the clinical impression. He was prescribed minocycline for scrub typhus. Second, facing the newly onset atrial fibrillation with rapid ventricular rate, consensus was needed whether the prescription of an anticoagulation agent or not. At that moment, cardiologists and neurologists were quick to estimate that the CHA2DS2-VASc^5^ score was one, meaning that the likelihood of a thromboembolic event such as a cerebral infarction was low. With the consent of the patient’s wife, we adopted a strategy of prioritizing potential infection control and delaying anticoagulant prescribing. Improvement in the liver, kidney, and DIC on the following day after admission to ICU (AST 61 U/L, ALT 48 U/L, BUN 19 mg/dL, Cr 0.8 mg/dL, D-dimer 9131.5 ng/mL) meant that the patient’s response to the prescribed antibiotics was good. He was transferred out of the intensive care unit on the third day of hospitalization and his vital signs were stable except for arrhythmia caused by atrial fibrillation. Scrub typhus infection was confirmed by serological tests. Weil-Felix test, OXK (1:20) was positive. On the fourth day of admission, he was prescribed a novel oral anticoagulant for persistent atrial fibrillation at the recommendation of a cardiologist due to improved kidney, liver function, and DIC. On the fifth day of hospitalization, the heart rhythm returned to normal sinus rhythm spontaneously. Finally, laboratory data showed no growth in blood cultures, and PCR analysis for O. tsutsugamushi was positive.

This patient was successfully discharged without any clinical sequela.

## 3. Discussion

Early effective treatment leads to better outcomes with shorter hospitalization stays and lowers the fatality rate [3]. The successful treatment of fatal septic shock caused by scrub typhus is mainly due to two key points: the first is early suspicion of the diagnosis, and the second is prescription with caution. These two key points will likely be difficult to reach without an immediate TRM conference for a full discussion among clinical specialists. More than 70 percent of serious medical errors result from problems with decision making, leadership, communication, teamwork, and situational awareness, so-called non-technical skills, which are preventable [6]. Our case report shows that in this severe case of multiple organ dysfunction and new-onset arrhythmia, the hospital should immediately organize a team of clinical specialists to assume responsibility for diagnosis and treatment rather than relying singling on a patient’s attending physician. The case report showed that the team of clinical specialists worked more efficiently in treating this patient. Following the TRM conference, the patient was identified with eschar and the clinical impression was established. Whether or not to prescribe anticoagulants to prevent thromboembolic events such as CVA in patients with new-onset atrial fibrillation and septic shock with DIC was also fully discussed. We adopted the CHA2DS2-VASc score as our scientific quantified tool for decision-making. The risk of thromboembolic events in guiding antithrombotic therapy in patients with atrial fibrillation or flutter was based on the assessment of the CHADS2-VASc score including congestive heart failure, hypertension, type 2 diabetes, age older than seventy-five years old, previous stroke or TIA, vascular disease and sex category [5]. The score of this patient was estimated as one which means a lower risk for thromboembolic episodes. We decided to defer the prescribing of anticoagulants because if the physician-patient was prescribed anticoagulants to prevent thromboembolism and accidentally developed ICH (intracranial hemorrhage), this would be unacceptable. Some serious complications of scrub typhus such as DIC, acute kidney injury, acute liver injury, pneumonia, gastrointestinal bleeding, myocarditis, and intracranial hemorrhage are common and may be fatal [7]. Fortunately, the patient responded well to minocycline, and the anticoagulant was used with caution without any complications.

Only with correct diagnosis and treatment and by avoiding complications can patients recover and be discharged without clinical sequelae. When the disease has high complexity and a worsening trend, there is a potential medical dispute crisis; the patient is a VIP or a colleague, the senior executive of the hospital, the attending physician, or the head of nursing takes the initiative to request, etc. would be an appropriate time to hold a TRM conference. In modern medical care, the timely implementation of a TRM conferences with invited family members, can effectively prevent mistakes and improve the quality of medical care through the multidisciplinary intervention of team members and the consensus on diagnosis and therapy reached after in-depth and extensive discussions [8,9].

## 4. Conclusions

The most significant difference between the TRM conference and the clinical specialist’s consultation is that it can quickly reach a consensus on the patient’s therapy strategy among doctors and family members, and effectively share the stress of the patient’s attending physician. The participation of family members is helpful for the investigation of the patient’s medical history, which is of great benefit to the diagnosis of the disease. Through multi-faceted and in-depth discussions and explanations, the medical team and family members will reach a consensus on the therapy procedure and then build mutual trust, and narrow the cognitive gap in treatment results, which can truly avoid medical disputes and achieve a win-win situation.

## Figures and Tables

**Figure 1 ijerph-19-10683-f001:**
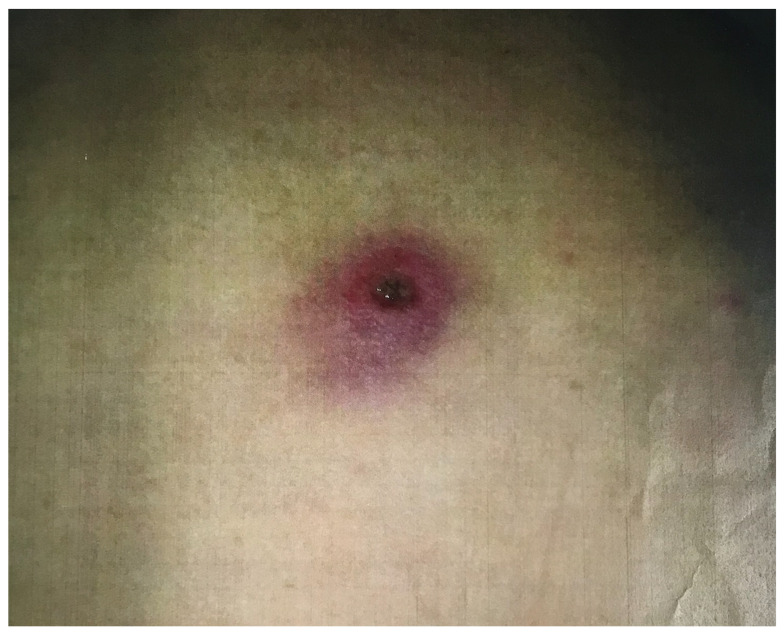
Skin lesion with the center developed a black crust covering the ulcer surrounding by an erythematous rim over the right buttock region.

## Data Availability

The data that support the findings of this case study are available from the corresponding author upon reasonable request.

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
