# Peer review of "Successful Application of Team Resource Management in Scrub Typhus Infection with Septic Shock"

_ijerph, 2022, doi:10.3390/ijerph191710683_

Round 1
Reviewer 1 Report
Thank you for the opportunity to review this case report. The report discusses a very interesting case in which good organizing, i.e. the use of a Team Resource Management (TRM) conference, literally turned out to be of vital importance for the patient. As far as I can judge, the manuscript meets the case report requirements of providing the requested information of an individual patient. Moreover, although the necessary medical jargon might distract the non-medically trained reader a bit, the report is fascinating. I have two issues for you to consider.
First, TRM conferences are often very expensive because of the large number of specialists involved. While the report demonstrates the potential value of TRM conferences, any reflections on the conditions under which to use or not to use TRM conferences could further strengthen the report.
Second, the abstract put much emphasis on the involvement of family members in TRM conferences, in terms resembling arguments from co-creation literature (see e.g. (Osborne, Radnor, & Strokosch, Co-Production and the Co-Creation of Value in Public Services: A suitable case for treatment?, 2016). However, the importance of family member involvement doesn’t show that clearly from the remainder of the manuscript. Please elaborate in more detail what family member involvement contributed to the diagnosis, treatment, and outcome.
Reviewer 2 Report
This article discusses the usefulness of Team resource management (TRM) conference for a patient with a severe infection.
The article is complete and well structured. English language should be improved: some sentences are difficult to understand or incomplete. Here are my comments:
Line 18: The full signification of TRM should appear on this line.
Line 45: The full signification of TRM should appear on this line.
Line 46: Correct the sentence «to attend the conference which is a significant difference»
Line 56: Correct the sentence «he was lucid but had generalized cold sweating»
Line 58: Correct the sentence «His heart sounds were normal»
Lines 68-69: Start a new sentence with the results of tests for infectious agents «Dengue Virus Ab IgM (-), Dengue virus 68 Ab IgG (-), Dengue NS1 Ag test (-), Influenza A Ag(-), Influenza B Ag(-).»
Line 73: The following acronyms are not explained: «LV systolic function, mild MR, mild TR, mild PR»
Line 79: Correct the sentence «with oxygen supplied through a nasal cannula»
Line 86: Correct the sentence «another thorough physical examination was performed on the patient»
Line 89: Correct the sentence «the specific feature of a wound due to a mite bite was highly suspected by a dermatologist»
Line 92: Correct the sentence «consensus was needed whether the prescription»
Lines 98-100: Correct the sentence «admission to ICU (AST 61U/L, ALT 48U/L, BUN 19mg/dL, Cr 98 0.8mg/dL, D-dimer 9131.5ng/mL) meant that the patient's response to the prescribed antibiotics was adequate»
Line 106: Correct the sentence «returned to normal sinus rhythm spontaneously»
Line 112: Correct the sentence «the first is early suspicion of the diagnosis»
Line 115: Correct the sentence «serious medical errors result from problems with decision making»
Line 125: Correct the sentence «with DIC was also fully discussed»
Lines 126-127: Correct the sentence «The risk of thromboembolic events in guiding antithrombotic therapy in patients with atrial fibrillation or flutter was based on the assessment»
Lines 131-133: Correct the sentence «We decided to defer the prescribing of anticoagulants because if the physician-patient was prescribed anticoagulants and accidentally developed ICH, this would be unacceptable»
Explain the acronym ICH.
Line 138: Correct the sentence «Only with correct diagnosis and treatment and by avoiding complications»
Lines 139-143: Correct the sentence « In modern medical care, the timely implementation of team resource management conferences with invited family members can effectively prevent mistakes and improve the quality of care through the multidisciplinary intervention of team members and the consensus on diagnosis and treatment reached after in-depth and extensive discussions»
General comment: It would be interesting for the authors to comment on the frequency of TRM conferences in their clinicals settings. I also wonder if the fact that the patient is a physician was an incentive to useTRM conference rapidly in this case.
